# An Emerging Strategy for Muscle Evanescent Trauma Discrimination by Spectroscopy and Chemometrics

**DOI:** 10.3390/ijms232113489

**Published:** 2022-11-04

**Authors:** Gongji Wang, Hao Wu, Canyu Yang, Zefeng Li, Run Chen, Xinggong Liang, Kai Yu, Huiyu Li, Chen Shen, Ruina Liu, Xin Wei, Qinru Sun, Kai Zhang, Zhenyuan Wang

**Affiliations:** Department of Forensic Pathology, College of Forensic Medicine, Xi’an Jiaotong University, Xi’an 710061, China

**Keywords:** evanescent trauma, ATR-FTIR, micro-IR spectroscopy, chemometrics

## Abstract

Trauma is one of the most common conditions in the biomedical field. It is important to identify it quickly and accurately. However, when evanescent trauma occurs, it presents a great challenge to professionals. There are few reports on the establishment of a rapid and accurate trauma identification and prediction model. In this study, Fourier transform infrared spectroscopy (FTIR) and microscopic spectroscopy (micro-IR) combined with chemometrics were used to establish prediction models for the rapid identification of muscle trauma in humans and rats. The results of the average spectrum, principal component analysis (PCA) and loading maps showed that the differences between the rat muscle trauma group and the rat control group were mainly related to biological macromolecules, such as proteins, nucleic acids and carbohydrates. The differences between the human muscle trauma group and the human control group were mainly related to proteins, polysaccharides, phospholipids and phosphates. Then, a partial least squares discriminant analysis (PLS-DA) was used to evaluate the classification ability of the training and test datasets. The classification accuracies were 99.10% and 93.69%, respectively. Moreover, a trauma classification and recognition model of human muscle tissue was constructed, and a good classification effect was obtained. The classification accuracies were 99.52% and 91.95%. In conclusion, spectroscopy and stoichiometry have the advantages of being rapid, accurate and objective and of having high resolution and a strong recognition ability, and they are emerging strategies for the identification of evanescent trauma. In addition, the combination of spectroscopy and stoichiometry has great potential in the application of medicine and criminal law under practical conditions.

## 1. Introduction

In recent years, the accurate judgment of trauma has occupied an important position in biomedical research [1,2]. In fact, not only can it provide direction for the treatment of patients with trauma in clinical medicine [1,2,3,4], but it can also provide clear clues and evidence for criminal review in terms of biological evidence [5,6]. In the process of forensic inspection, there is a phenomenon that has long been a problem in the judicial personnel; i.e., when the human body has permanently ceased its bodily functions and lost all vital activities, under the joint action of irreversible tissue autolysis and external bacteria, the tissue form of the body disappears and enters a state of disorder, and the possible trauma is buried. We originally named this form of trauma “evanescent trauma”. When evanescent trauma occurs, the traditional morphological methods that have been relied on can no longer meet the needs of diagnosis, which causes great frustration and forms obstacles to judicial personnel. Removing the shackles of traditional methods and identifying new methods may solve evanescent trauma discrimination.

From the 1990s to the present time, vibration spectroscopy has made great progress in the field of biological fluid analysis, showing its advantages of accuracy, speed and user friendliness [7,8,9]. Some studies have shown that the use of Fourier transform infrared spectroscopy combined with chemometrics obtains good classification results for bloodstains [10,11,12,13,14], semen spots [15,16,17], pulmonary edema fluid [15,18,19,20] and saliva [15,21,22,23]. Cai et al. [24] used a combination of infrared spectroscopy and chemometrics to analyze and discriminate brain tissue damage in mice during postmortem examination. Zhang et al. [25] used microscopic infrared technology to perform a spectral imaging analysis of rat postmortem muscle tissue, classified and discriminated red blood cell hemoglobin and proposed that the double-concave disk shape of red blood cells disappeared after 4 days of postmortem placement of rats. All these studies show the advantages of infrared spectroscopy and stoichiometry in the field of biological evidence. However, there is no comprehensive and accurate judgment for discriminant analyses of evanescent trauma.

Therefore, the focus of this study was to develop a broader, more accurate, and more systematic technique for identifying and analyzing muscle evanescent trauma using attenuated total reflection (ATR) Fourier transform infrared spectroscopy (FTIR), as well as microscopic infrared spectroscopy (micro-IR) combined with appropriate chemometrics. This method integrates fresh and evanescent trauma to build a more complete trauma recognition model. In addition, human muscle samples were also used in this study to establish the trauma recognition model, which is more in line with the actual needs.

## 2. Results and Discussion

### 2.1. Macroscopic and Microscopic Visualization

In traditional trauma identification and judgment, macroscopic observation and microscopic HE staining are commonly used, and the gold index of muscle tissue trauma is determined; that is, muscle cells are broken irregularly, and red blood cells can be seen in the intercellular space. In this study, the macroscopic and microscopic images of the trauma group and the control group at different times, that is, the macroscopic observation and HE staining, were compared and displayed. HE staining is shown in Figure 1, and the macroscopic observation is shown in Appendix A. The macroscopic observation showed that, in the early stages of injury (0d and 1d), muscle contusion and massive bleeding occurred, and after 5d, there was a large amount of corrupt fluid, dried tissue and shapeless tissue. In the control group, except the early bleeding of the injury, a similar phenomenon of decay decomposition could be seen.

The results of the HE staining showed that, at the early stages of injury, that is, at 0d and 1d, there was a bleeding area, a large number of red blood cells in the space of the muscle tissue, broken muscle fibers and clearly visible muscle nuclei. At 5d, only a vague outline was seen in the morphology of the myocytes, no red blood cells were seen in the interstitial space, and the nuclei of the myocytes disappeared. At 10d and 15d, the normal morphology of the myocytes disappeared under the microscope, turning into homogeneous red staining or even disorder. This may be due to the loss of all life activities of the body, such as that of intracellular lysosomes; that is, intracellular organs containing a series of proteolytic enzymes and other digestive enzymes are digested by their own enzymes, causing organelle membrane rupture, releasing a large number of digestive enzymes into the muscle cells and burying evidence that can identify muscle cell damage. At the same time, we thought there might be another reason for this, whereby the decomposition of external bacteria was also involved. We refer to this form of trauma as “evanescent trauma”, in which the trauma to the tissue becomes unidentifiable or even undiagnosable.

### 2.2. Spectral Characteristic Peaks and Changes in Time Sequence

In this study, we preliminarily judged and analyzed the mean spectrogram between the trauma group and the control group. Some of the spectral peaks of the focus studies are recorded in Table 1. As shown in Figure 2a, the peak spectral differences between the trauma group and the control group are mainly concentrated in the regions of 1700–1500 cm^−1^, 1500–1200 cm^−1^, 1200–900 cm^−1^ and others. These major peak differences may suggest that the difference in contribution between the trauma group and the control group depends on the Amide I band, Amide II band, nucleic acids, carbohydrates and others. In order to visually display the spectral differences in this region, the focus of this study was adjusted to the spectral range of 1800–900 cm^−1^, namely, the biological fingerprint region [26], which provides the maximum information of compounds in biological samples. Figure 2b shows that the main differences occurred in the four regions of 1100–1040 cm^−1^, 1402 cm^−1^, 1570–1510 cm^−1^ and 1695–1620 cm^−1^. Among them, 1695–1620 cm^−1^ is due to the unshared electron pair on the nitrogen atom and the p-π conjugate of the carbonyl group, so the ν_C=O_ stretching frequency decreases, indicating the Amide I band; 1570–1510 cm^−1^ is due to the in-plane deformation vibration of NH_2_, indicating the Amide II band; 1402 cm^−1^ is the symmetric CH_3_ bending modes of the methyl groups of proteins; and 1100–1040 cm^−1^ is the stretching of the P=O symmetric of the >PO_2_^−^ groups of nucleic acids, hinting at phospholipids.

In this study, the difference in the mean spectra between the trauma group and the control group at different times was also noted, as shown in Figure 2c,d. We found that the above peaks changed with time. For example, in the trauma group, the peak absorbance of the Amide Ⅰ band (1695–1620 cm^−1^) was in the following order: 1d > 0d > 5d > 10d > 15d; the peak absorbance of the Amide Ⅱ band (1570–1510 cm^−1^) was in the following order: 15d > 5d > 10d > 1d > 0d; and the order of the absorption peak of the symmetric CH_3_ bend (1402 cm^−1^) was as follows: 10d > 15d > 5d > 1d > 0d. In the control group, it was found that the peak order of the Amide Ⅰ band (1695–1620 cm^−1^) was consistent with that of the trauma group. The peak absorbance of the Amide Ⅱ band (1570–1510 cm^−1^) was in the following order: 15d > 10d > 5d > 1d > 0d. The order of the peak absorbance of the symmetric CH_3_ bend (1402 cm^−1^) was in the following order: 15d > 10d > 5d > 1d > 0d. This shows that, as time progresses, the trauma factor is not completely masked, and the differences in some groups can still be determined using spectroscopy. 

### 2.3. Classification Model of Rat Muscle Trauma Based on ATR-FTIR

In this study, PCA, an unsupervised discrimination method, was used for analyses. As can be seen in Figure 3a, the distribution of the samples in the trauma group was uniform and scattered, while that in the control group was concentrated, mainly distributed on the left, upper right and lower right of the figure. The PCA analysis results show that the interpretation variation in the spectral data score plots of the trauma group and the control group was about 97% on PC1 and PC2, and we speculate that the reason why the two groups could not be completely separated might be the placement time. However, there was no coincidence between the trauma group and the control group, indicating that there were differences between the two groups. In the loading map of PC1, we noticed some significant load absorption peaks, which might play an important role in explaining the differences between the two groups (Figure 3b). Among them, 1714 cm^−1^ is C=O thymine, suggesting that nucleic acid may be involved; 1560 cm^−1^ is Amide Ⅱ, suggesting that proteins may be involved; 1466 cm^−1^ is the CH_2_ scissoring mode of the acyl chain of lipid, the cholesterol-methyl band, and 1402 cm^−1^ is the symmetric CH_3_ bending modes of the methyl groups of proteins, suggesting that proteins may be involved; and 1010 cm^−1^ is a stretch C-O, suggesting that carbohydrates may be involved. In general, the differences between the trauma group and the control group might be related to proteins, lipids, nucleic acids, carbohydrates and their metabolites.

In order to identify the observed values between the groups and the influencing factors that may lead to the differences between the groups, PLS-DA was selected as a classification technique to establish a trauma classification model of rat muscle tissue based on ATR-FTIR. Eight LVs were selected, and the PLS-DA model was constructed based on the average value of the minimum CV and Cal classification errors. In Appendix A, we found a certain trend of separation between the trauma group and the control group in the LV1 and LV2 directions, which, at the same time, corroborated the results in Figure 3a. In Figure 3c, the red dashed line indicates whether the sample is the trauma or control group, the red mark above the threshold line is the trauma group, and the green mark below the threshold is the control group. The sensitivity and specificity of the classification of rat trauma were 0.958 and 0.970, respectively. As shown in the ROC curve in Appendix A, the accuracy of the classification model was 99.10%. This shows that the PLS-DA classification model is a robust model.

To further test the predictive power of this supervised classification model, we loaded the externally validated datasets into the PLS-DA model. As shown in Figure 3d, the dashed red line is the border. The red diamond above the border is the trauma group, and the green rectangle below the border is the negative control group. The prediction results showed that the classification results of the test dataset were all 0.852 (sensitivity and specificity), and the accuracy rate was 93.69% (Appendix A), which also indicates the robustness and usability of the muscle trauma classification model. 

### 2.4. Classification Model of Human Muscle Trauma Based on Micro-IR

In order to be more in line with the actual situation, the PCA analysis was performed on the datasets collected using micro-infrared spectroscopy, and the results are shown in Figure 4. In the direction of PC1 and PC2, the difference between the trauma group and the control group was explained by about 72.5%. The distribution of the scores between the trauma group and the control group showed a certain regularity; that is, according to the trend from concentration to dispersion, the distribution of the trauma group marked with green was from the bottom right to the top left, while the distribution of the control group marked with red was from the bottom left to the top right (Figure 4a). To some extent, there was a tendency for the two groups to separate, and the differences may be concentrated in PC1 and PC2. In the load diagram, in the direction of PC1 (Figure 4b), we noticed 1662 cm^−1^, 1557 cm^−1^, 1338 cm^−1^, 1239 cm^−1^ and 1201 cm^−1^. Among them, 1662 cm^−1^ and 1557 cm^−1^ are Amide Ⅰ and Amide Ⅱ, respectively; 1338 cm^−1^ is CH_2_ wagging; 1239 cm^−1^ is asymmetric PO_2_^−^ stretching; and 1201 cm^−1^ is PO_2_^−^ asymmetric, phosphate I. In the load diagram, in the direction of PC2 (Figure 4c), we noticed 1738 cm^−1^, 1449 cm^−1^, 1243 cm^−1^ and 1089 cm^−1^. Among them, 1738 cm^−1^ is ν(C=O), polysaccharides and hemicellulose; 1449 cm^−1^ is the asymmetric CH_3_ bending of the methyl groups of proteins; 1243 cm^−1^ is the ν(PO_2_^−^) asymmetric stretching of phosphodiesters; and 1089 cm^−1^ is the stretching of the P=O symmetric of the >PO_2_^−^ groups of nucleic acids and phospholipids. In summary, proteins, polysaccharides, phospholipids and phosphates might be related to the main differences between the trauma and control groups in human muscle samples.

Similarly, PLS-DA was selected to determine the factors influencing the differences between the groups, and at the same time, a human muscle tissue trauma classification model was established based on micro-IR. Six LVs were selected to construct the PLS-DA model. In Appendix A, we found a certain trend of separation between the positive and negative groups in the LV1 and LV2 directions, which, at the same time, corroborated the results in Figure 4a. In Figure 4d, the red dashed line indicates whether the sample is the trauma or control group, the green above the threshold line is labeled as the trauma group (Pos), and the green below the threshold is labeled as the control group (Neg). The sensitivity and specificity of the human muscle trauma classification were 0.962 and 0.970, respectively. As shown in the ROC curve of Appendix A, the accuracy of the classification model was 99.52%.

To further test the predictive ability of the supervised classification model, externally validated datasets were loaded into the PLS-DA model. As shown in Figure 4e, the dashed red line is the border. The green mark above the border is the trauma group, and the red mark below the border is the negative control group. The prediction results showed that the classification results of the test dataset were 0.868 and 0.903 (sensitivity and specificity, respectively), and the accuracy was 91.95% (Appendix A), which also indicates the robustness of the muscle injury classification model.

## 3. Materials and Methods

### 3.1. Animal Model Establishment and Sample Preparation

This research was approved by the Laboratory Animal Care Committee of Xi’an Jiaotong University and complied with the recommendations in the Xi’an Jiaotong University Guide for the Care and Use of Laboratory Animals. In this study, 120 SPF healthy adult male Sprague Dawley (SD) rats, weighing 220–260 g, were purchased from the Laboratory Animal Center of Xi’an Jiaotong University. All rat models of muscle trauma were performed under 2% isoflurane air anesthesia.

A total of 120 adult SD male rats were randomly divided into a muscle trauma group and a negative control group, with 60 rats in each group. Before the experiment, the animals were adapted to the modeling room for 1 h and provided with an adequate diet. After the successful induction of anesthesia, closed contusion of the left lower limb gastrocnemius muscle was induced by free-falling percussion (conditions of self-made strike device: weight 340 g). After successful modeling, the rats were fed normally for 1 h, and then all the rats were sacrificed by cervical dislocation. After the rats were sacrificed, the trauma group and the negative control group were placed in an incubator (temperature: 25 ± 1 °C; humidity: 50 ± 5%) for 0d, 1d, 5d, 10d and 15d, with 12 rats in each group. Among them, 10 rats were used for ATR-FTIR spectroscopy detection, and the other 2 rats were used for hematoxylin and eosin staining (HE), that is, macroscopic and microscopic visualization. A total of 220 muscle samples were collected at the corresponding time points, among which 100 samples were used to collect spectral data, 20 samples were used for HE staining, and the remaining 100 samples were stored in an ultra-low temperature refrigerator at −80 °C for repeated experiments.

In addition, we also collected 32 human muscle samples, including 16 cases of trauma muscle tissue and 16 cases of negative muscle tissue from the same individual. Among them, 8 cases of trauma muscle tissue and 8 cases of negative muscle tissue were cultured in vitro in a constant temperature and humidity chamber for 4 days, and all samples were used to collect spectral data. It should be noted that the rat muscle samples used for ATR-FTIR infrared detection need to be homogenized, and the human muscle samples for microscopic infrared spectroscopy were processed using a freezing microtome. The cut sections were deposited on infrared transparent calcium fluoride (CaF_2_) slides for spectral collection. The remaining samples were stored at −80 °C for repeated testing. It should be emphasized that the muscle samples were collected in accordance with the guidelines of the Laboratory Animal Care Committee of Xi’an Jiaotong University and with informed written consent from the immediate family of each donor and the relevant institutions.

### 3.2. Spectral Collection and Data Preprocessing

Spectral data from the rat samples were acquired using a Nicolet 5700 FTIR spectrometer (Thermo Fisher Scientific, Waltham, WA, USA), and a diamond crystal ATR (Thermo Fisher Scientific, Waltham, WA, USA) was equipped for spectral acquisition. After each background collection, 1 uL of the prepared muscle homogenate sample was added to the spectral probe drop wise. Spectra were collected over a range of 900 to 4000 cm^−1^, with a resolution of 4 cm^−1^ and 32 scans. In order to minimize errors caused by unevenly added samples, each sample was analyzed 9 times (each analysis of 9 repeated spectra) and then averaged to form a spectrum representing one sample. Finally, a total of 900 spectral data were obtained by ATR infrared spectrum detection. Spectra were recorded using OMNIC software version 8.0 (Thermo Fisher Scientific, Waltham, WA, USA).

Spectral data from the human samples were collected using a Varian 660-IR spectrometer coupled to a Varian 620-IR spectrometer imaging microscope (Agilent Technologies, CA, USA). A liquid-nitrogen-cooled mercury-cadmium-telluride (MCT) focal-plane array (FPA) detector consists of 4096 pixels arranged in a 64 × 64 grid format. Spectra were collected between 3950 and 950 cm^−1^, with a spectral resolution of 4 cm^−1^ for 32 sample spectral scans and 64 background spectral scans. The background spectrum was selected as a blank area of each CaF_2_ slide, and it was automatically subtracted from each spectrum before the sample tissue spectrum was taken. Each tissue section with an area of 352 × 352 μm^2^ sampled from a typical tissue area was scanned at a pixel resolution of 5.5 × 5.5 μm^2^, which contained 4096 spectra. After that, quality tests were assigned to each infrared image, including sample thickness tests and signal-to-noise ratio tests. In the sample thickness test, spectra from regions with little or no tissue and too-thick tissue were discarded. The upper and lower thresholds were defined as 1.6 and 0.2, respectively.

Next, the rat muscle spectrum and human muscle spectrum were cut into the 1800–900 cm^−1^ region, namely, the biometric fingerprint region. After that, Savitzky–Golay (SG) smoothing was used, and standard normal variables (SNVs) were applied to selected spectral regions in order to reduce the effects of light scattering and sample thickness. Finally, baseline correction and average center correction were performed to solve the spectral overlap problem [27].

In addition, it should be noted that 70% of the rat muscle data detected by ATR-FTIR were used for modeling, and the remaining 30% were used for external validation. Similarly, of the human sample data examined by microscopic infrared spectroscopy, 75% were used for modeling, and the remaining 25% were used for external validation.

### 3.3. Multivariable Statistical Analysis and Software

In the current field of chemometrics, pattern recognition mainly consists of unsupervised and supervised patterns [27]. The unsupervised pattern, which groups data structures but does not provide training criteria, is implemented algorithmically. The method of classifying and identifying new unknown samples by using the information contained in the samples is called the supervised method. In this study, the unsupervised and the supervised methods were used to analyze the spectral data of the rat muscle and human muscle, respectively.

The principal component analysis (PCA) used in this study is a common algorithm used to reduce dataset dimensions. By searching the orthogonal direction with the largest dispersion, the distribution of each sample is identified in the multidimensional space of the original variable. Linear models are built based on this rule, and principal components (PCs) can be extracted from the original data. In addition, this recognition mode can retain the main information in the original data. In simple terms, this model uses the projection of the sample on a given PC, shows the difference in the data through a score map and combines the load map with the corresponding score map to judge the contribution degree of the variable to the total variation [27,28]. In this study, the principal component analysis was used to analyze the muscle samples with different placement times so as to determine the contribution of trauma as a variable to the total variation and to provide a new strategy for identifying muscle evanescent trauma.

However, since PCA is an unsupervised analysis method, it cannot distinguish the contribution of each individual sample from that of all samples to the model. If the differences between samples are large and the differences within groups are small, this method can clearly distinguish the differences between samples. However, if the differences between groups are small and the differences within groups are large, it is difficult for PCA to detect and distinguish the differences between groups. In addition, if the differences between groups are small and the sample size of each group is large, the group with the larger sample size will play a major role in influencing the model [27,28,29,30].

In summary, the supervised categorical discriminant method chosen in this study was the partial least squares discriminant analysis (PLS-DA), which can solve these problems in PCA. Similarly, it is an analysis method that is frequently used in research to handle the classification and discrimination of multivariate data. The idea of the PLS method is to construct orthogonal score vectors (latent variables or principal components) by maximizing the covariance between the independent variable data and the respondent dataset so as to fit the linear relationship between the independent variable data and the respondent data. The difference between PLS and PCA is that PLS decomposes both the independent variable X matrix and the respondent variable Y matrix, and it uses covariance information in the decomposition, which enables the dimensionality reduction effect to extract intergroup variation information more efficiently than PCA. 

All spectral data were analyzed using MATLAB R2020b (The MathWorks, Natick, MA, USA) and PLS Toolbox 7.9 (Eigenvector Research, Wenatchee, WA, USA).

## 4. Conclusions

Preliminary studies have shown that the combination of FTIR spectroscopy, microscopic infrared spectroscopy and advanced stoichiometry is an accurate and novel strategy for determining vanishing muscle trauma, especially in realistic forensic practice. In this study, the differences between the trauma group and the control group, as well as the changes in the time-series data of the two groups, were analyzed by means of an average spectrum of rat muscle, and it was concluded that the differences between the two groups might lie in the proteins, nucleic acids, carbohydrates and others. The differences between the proteins, lipids, nucleic acids, nucleic acids and their metabolites were determined by PCA. The result is basically consistent with the conclusion obtained from the average spectrum. Based on the PLS-DA algorithm and the training dataset comprising 70% of the rat muscle samples, a rat muscle trauma recognition model was established. The sensitivity and specificity of the model were 0.958 and 0.970, respectively. The accuracy was 99.10%. We then verified the robustness of the model using an external test dataset that accounted for 30% of the total data. The classification results of the model were 0.852 and 0.852 (sensitivity and specificity, respectively). The accuracy was 93.69%. 

At the same time, an identification model of the human muscle trauma recognition model was established. According to the results of the PCA score map and the load map, we believed that proteins, polysaccharides, phospholipids and phosphates might be related to the main differences between the trauma and control groups in human muscle samples. In addition, the classification results of the model were 0.962 and 0.970 (sensitivity and specificity, respectively), and the accuracy was 99.52%. The classification results of the external test dataset were 0.868 and 0.903 (sensitivity and specificity, respectively), and the accuracy was 91.95%. The PLS-DA classification model is robust and fits well, and it has good application prospects in forensic practice.

However, the situation encountered in actual judicial practice is much more complex. This study only considers the samples placed under constant temperature and humidity conditions, without considering the influence of environmental factors (such as humidity). At the same time, the differences between the trauma group and the control group were only limited to biological macromolecules, and the specific molecules and their contents and effects on the body were not explored. In addition, the small sample size of human muscle samples and whether the gender of the clinical samples influenced the results were not studied in further targeted analyses; this was also the case for the animal samples, and no female rats were introduced, which is a limitation of this study. Therefore, these factors should be analyzed in subsequent studies so that they can be applied in medical practice as soon as possible.

## Figures and Tables

**Figure 1 ijms-23-13489-f001:**
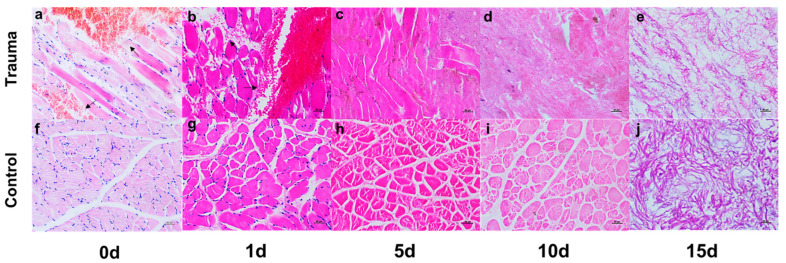
Microscopic visualization of HE staining in trauma group and control group at different time points. (**a**–**e**) The observation of the wound group under the HE staining microscope. The rose red color shows the cytoplasm and intercellular stroma, blue shows the nucleus, and the black arrow indicates the bleeding area and red blood cells. (**f**–**j**) The observation of the control group under the microscope of HE staining. Rose red is the cytoplasm and intercellular stroma, and blue is the nucleus.

**Figure 2 ijms-23-13489-f002:**
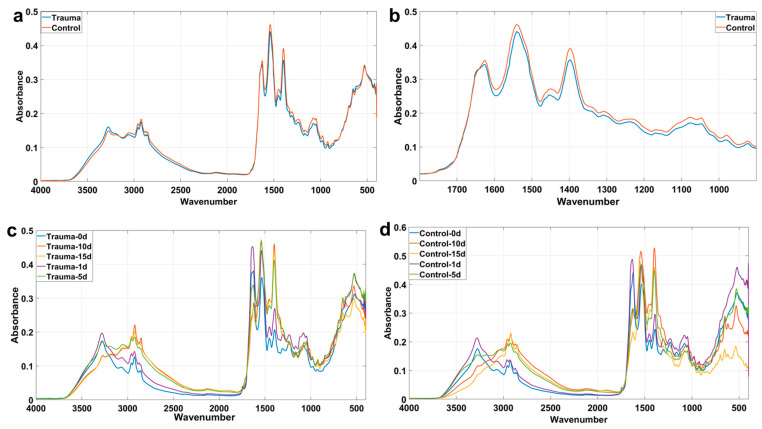
Average spectrogram and peak diagram of biological fingerprint area of trauma group and control group. (**a**) Mean spectrogram of trauma group and control group; (**b**) biometric fingerprint region of the mean spectrogram of the trauma group and the control group; (**c**) mean spectrogram of trauma group at different times; (**d**) mean spectrogram of the control group at different times.

**Figure 3 ijms-23-13489-f003:**
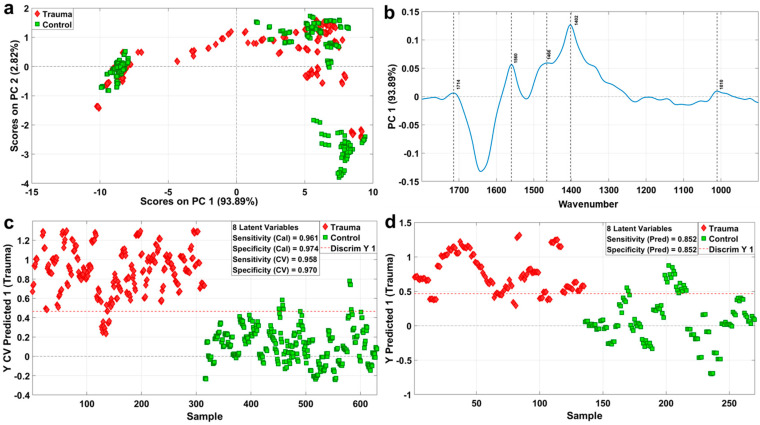
Classification model of rat muscle trauma and its accuracy index. (**a**) PCA scores in the direction of PC1 and PC2 of the trauma group and the control group. The red markers are trauma group, and the green markers are control group. (**b**) Loading plot on PC1 from PCA analysis, marked with significant peaks. (**c**) Trauma classification model and sensitivity and specificity in-dexes established based on training sets of trauma group and control group. The red markers are trauma group, and the green markers are control group. (**d**) Trauma classification prediction model and related evaluation model indexes established based on the test sets of trauma group and control group. The red markers are trauma group, and the green markers are control group.

**Figure 4 ijms-23-13489-f004:**
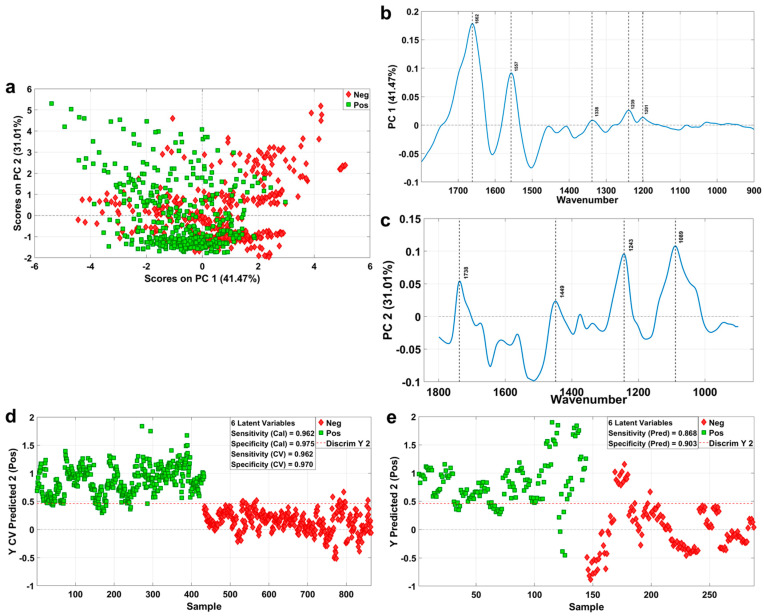
Classification model of human muscle trauma and its accuracy index. (**a**) PCA scores in the direction of PC1 and PC2 of the trauma group and the control group. The green markers are trauma-positive group, and the red markers are negative control group. (**b**) Loading plot on PC1 from PCA analysis, marked with significant peaks. (**c**) Loading plot on PC2 from PCA analysis, marked with significant peaks. (**d**) Trauma classification model and sensitivity and specificity in-dexes established based on training sets of trauma group and control group. The green markers are trauma-positive group, and the red markers are negative control group. (**e**) Trauma classification prediction model and related evaluation model indexes established based on the test sets of trauma group and control group. The green markers are trauma-positive group, and the red markers are negative control group.

**Table 1 ijms-23-13489-t001:** Peak component assignment.

Wavenumber (cm^−1^)	Assignment
~1011–1009	Stretch C-O, carbohydrates
~1100–1040	Stretch P=O symmetric of the >PO_2_^−^ groups of nucleic acids, phospholipids
~1201	PO_2_^−^ asymmetric, phosphate I
~1239–1238	Asymmetric PO_2_^−^ stretching
~1243	ν(PO_2_^−^) asymmetric stretching of phosphodiesters
~1338–1337	CH_2_ wagging
~1402	Symmetric CH_3_ bending modes of the methyl groups of proteins
~1449	Asymmetric CH_3_ bending of the methyl groups of proteins
~1467–1465	CH_2_ scissoring mode of the acyl chain of lipid, cholesterol-methyl band
~1570–1510	Amide II
~1695–1620	Amide Ⅰ
~1714	C=O thymine
~1739–1738	ν(C=O), polysaccharides, hemicellulose

## Data Availability

The data presented in this study are available in Appendix A.

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
