# Peer review of "An Emerging Strategy for Muscle Evanescent Trauma Discrimination by Spectroscopy and Chemometrics"

_ijms, 2022, doi:10.3390/ijms232113489_

Round 1
Reviewer 1 Report
This is an elegantly written manuscript, describing the strategy to discriminate muscle evanescent trauma by means of spectroscopy and chemometrics techniques. The work described is of high quality, and the manuscript is clear and well written. Here are some comments:–
Major changes
· Since this study only included male rats (and no female rats and the authors did not mentioned about the genders for the clinical samples) that makes it challenging to extract any vital conclusions. Therefore, I suggests the authors to add these limitations in their limitation section.
Minor changes
· Please follow up “et al” with period i.e. “et al.” throughout the manuscript.
· In line 199, use “at” instead of “At”.
· Please fix sub-script and super-script throughout the manuscript, including in tables and figures, i.e. write “CH2” instead of “CH2”, “CH3” instead of “CH3”, “PO2-” instead of “PO2-”.
· In line 213, write “cm-1” instead of “cm-1”.
· In line 265, please remove period after “stretch”.
· In Figure 2a and Figure 2b, the authors showed the mean spectrogram for the trauma group and the control group. Are these mean spectrogram belongs to 0d or 1d or 5d or 10d or 15d time point? Please clarify.
· In line 265, remove period after “stretch”.
· Please use “and more” instead of “etc” throughout the manuscript and make adjustment to the respected sentence according to that.
· Please make sure all references are according to IJMS (MDPI) format.
Author Response
Dear reviewer,
We greatly appreciate your positive comments on this research manuscript as well as your constructive and positive comments. Here is our point-by-point response according to the reviewers’ comments. In addition, we have included a revised manuscript with tracked changes.
If you have any questions about this manuscript, please do not hesitate to contact us.
Thank you so much.
Yours sincerely,
Corresponding author
Zhenyuan Wang
Reviewer(s)' Comments to Author and Responses:
Major change:
Since this study only included male rats (and no female rats and the authors did not mentioned about the genders for the clinical samples) that makes it challenging to extract any vital conclusions. Therefore, I suggests the authors to add these limitations in their limitation section.
Response:
Thank you for your positive comments, careful review and constructive suggestions. In the revised version of the manuscript, we have added the limitations you mentioned to the corresponding sections and also marked them.
Minor changes:
- Please follow up “et al” with period i.e. “et al.” throughout the manuscript.
Response:
Thanks for your constructive comment. We have modified in the revised manuscript carefully.
- In line 199, use “at” instead of “At”.
Response:
Thanks for your constructive comment. We have modified in the revised manuscript carefully. 3. Please fix sub-script and super-script throughout the manuscript, including in tables and figures, i.e. write “CH2” instead of “CH2”, “CH3” instead of “CH3”, “PO2-” instead of “PO2-”.
Response:
Thanks for your constructive comment. We have modified in the revised manuscript carefully.
- In line 213, write “cm-1” instead of “cm-1”.
Response:
Thanks for your constructive comment. We have modified in the revised manuscript carefully.
- In line 265, please remove period after “stretch”.
Response:
Thanks for your constructive comment. We have modified in the revised manuscript carefully.
- In Figure 2a and Figure 2b, the authors showed the mean spectrogram for the trauma group and the control group. Are these mean spectrogram belongs to 0d or 1d or 5d or 10d or 15d time point? Please clarify.
Response:
Thanks for your constructive comment. The mean spectrograms of the trauma and control groups mentioned in Figure 2a and Figure 2b were indeed obtained by averaging the entire sample at the five time points you mentioned. Since the focus of this paper is to identify differences between the trauma and control groups, we averaged all spectral data (including all time points) to observe the differences between the two groups on the average spectrogram.
- In line 265, remove period after “stretch”.
Response:
Thanks for your constructive comment. We have modified in the revised manuscript carefully.
- Please use “and more” instead of “etc” throughout the manuscript and make adjustment to the respected sentence according to that.
Response:
Thanks for your constructive comment. We have modified in the revised manuscript carefully.
- Please make sure all references are according to IJMS (MDPI) format.
Response:
Thanks for your constructive comment. All references have been carefully checked to ensure compliance with the IJMS (MDPI) format.

Reviewer 2 Report
The main idea of ​​the work is based on the use of chemometric methods (PCA and PLS-DA) to prove the usefulness of spectral data in the IR range for the identification and discrimination of muscle trauma in humans and rats. The study is interesting, but the presented results of chemometric analysis are not fully convincing. The authors should better justify that models are not overfitted.
1. The main benefit of PLS-DA model in analyzing spectral data is identifying spectral features responsible for separating groups, i.e experimental vs control. It seems that the plot scores (LV vs samples) from PLS-DA would be a better choice for detecting bands which discriminate both groups than the plot of PC vs samples from PCA (Fig. 3b and 4b). Please compare whether the score plots in PLS-DA analysis give conclusions consistent with those presented.
2. Calibration and cross-validation errors (RMSEC and RMSECV) for PCA analysis were not presented.
3. How was the sample set divided into test and validation subsets? For rat and human muscle samples, different proportions of samples in calibration and test sets were used (70% vs 30% and 75% vs 25%). Why?
4. If a PCA model fails to achieve group separation (as in the case of your study), the PLS-DA model is often unreliable or invalid despite the appearance of group separation. In case PLS-DA reveals group separations even when PCA does not, these results require rigorous validation to prove the model's reliability. Has the permutation test (for min 100 iterations) been performed?
5. The sentence “Through proper rotation transformation principal components, the influencing factors leading to the differences between sample groups were found “ is unclear. What do you mean by “proper rotation transformation”? Manually rotation of model loadings?
Author Response
Dear reviewer,
We greatly appreciate you for the constructive and professional comments. Here is our point-by-point response according to the reviewers’ comments. In addition, we have included a revised manuscript with tracked changes.
If you have any inquiries about this version, please do not hesitate to contact us.
Thank you so much.
Yours sincerely,
Corresponding author
Zhenyuan Wang
Comments and Suggestions for Authors and Responses:
The main idea of the work is based on the use of chemometric methods (PCA and PLS-DA) to prove the usefulness of spectral data in the IR range for the identification and discrimination of muscle trauma in humans and rats. The study is interesting, but the presented results of chemometric analysis are not fully convincing. The authors should better justify that models are not overfitted.
Response:
Thank you for your professional comments, careful review and constructive suggestions. As we all know, overfitting and underfitting are very important factors for classification recognition models, as you said, it is related to the scientific validity and usability of the study results. I will explain and revise the problems you pointed out and suggest changes on a point-by-point basis.
- The main benefit of PLS-DA model in analyzing spectral data is identifying spectral features responsible for separating groups, i.e experimental vs control. It seems that the plot scores (LV vs samples) from PLS-DA would be a better choice for detecting bands which discriminate both groups than the plot of PC vs samples from PCA (Fig. 3b and 4b). Please compare whether the score plots in PLS-DA analysis give conclusions consistent with those presented.
Response:
First of all, thank you for your professional comments. Next, in the manuscript we used PCA for analysis, in fact, only PCA was used for dimensionality reduction of high-dimensional data, and PCA was not used for modeling to identify trauma. Finally, we have supplemented the score plot of PLS-DA analysis as you suggested, and the figure is attached below and also added to the supplementary material of the revised version. According to the information in the figures, we can know that after PLS-DA analysis, there is a tendency to separate between the trauma group and the control group to some extent, although not completely in the LV1 and LV2 directions, and also corresponds to the results of PCA analysis in the manuscript. The same was found in the plots of human samples.
- Calibration and cross-validation errors (RMSEC and RMSECV) for PCA analysis were not presented.
First of all, thank you for your professional comments. Next, we have provided the calibration and cross-validation errors (RMSEC and RMSECV) for PCA analysis below and added them to the manuscript supplement. As shown in the figure, after the 8th principal component, we found that the RMSEC and RMSECV spacing gradually became larger, and in accordance with the early stopping principle to prevent the occurrence of overfitting, we selected 8 PCs. Similarly, in human samples, we selected 6PCs.
- How was the sample set divided into test and validation subsets? For rat and human muscle samples, different proportions of samples in calibration and test sets were used (70% vs 30% and 75% vs 25%). Why?
Response:
First of all, thank you very much for your professional questions. We randomly divided the total dataset into two parts. In the dataset of rats, since there are 10 rats in each group, the allocation was made according to the ratio of 7:3, according to the principle of single-blindness, 30% of which the external validation dataset is unknown to the data analysts. In contrast, in the dataset of human samples, there are only 16 samples in each group, therefore, to facilitate the allocation of the training set to the external validation set, 75%:25% was chosen between the conventional 8:2 and 7:3, and as above, this 25% of the dataset was unknown to the test analysts.
- If a PCA model fails to achieve group separation (as in the case of your study), the PLS-DA model is often unreliable or invalid despite the appearance of group separation. In case PLS-DA reveals group separations even when PCA does not, these results require rigorous validation to prove the model's reliability. Has the permutation test (for min 100 iterations) been performed?
Response:
First of all, thank you very much for your professional questions. In general, PCA results are often less than ideal due to the high-dimensional, small-sample nature of spectral data, as with data such as metabolomics, and the interference of noisy variables. Nevertheless, we can observe whether there is a trend of separation between groups and data outliers based on the score plot of PCA, and when there is some trend of separation between group classification, it indicates that there is a component between the two that can be classified. And PLS-DA, a supervised classification method, incorporates a regression model along with dimensionality reduction, and uses certain discriminant thresholds to perform discriminant analysis on the regression results. However, when using supervised learning methods for analysis, it is easy to produce overfitting, so we used the internal cross-validation of blinds and the most effective current model validation method, i.e., dividing the entire data set into two parts: internal training data and external validation data, using the internal training set for model building, and then using the external validation set for prediction, to objectively evaluate the validity and The validity and applicability of the model are evaluated objectively.
- The sentence “Through proper rotation transformation principal components, the influencing factors leading to the differences between sample groups were found” is unclear. What do you mean by “proper rotation transformation”? Manually rotation of model loadings?
Response:
First of all, thank you very much for your professional questions. In fact, the point you mentioned is actually based on the idea of PLS method, which is to construct orthogonal score vectors (latent variables or principal components) by maximizing the covariance between the independent variable data and the respondent data set, so as to fit the linear relationship between the independent variable data and the respondent data. The difference between PLS and PCA is that PLS decomposes both the independent variable X matrix and the respondent variable Y matrix, and uses covariance information in the decomposition, which enables the dimensionality reduction effect to extract intergroup variation information more efficiently than PCA. Appreciate your pointing out the problems, we have improved and added in the section of materials and methods.
